# Training Long Short-Term Memory with Sparsified Stochastic Gradient Descent

**Maohua Zhu, Yuan Xie**
Department of Electrical and Computer Engineering
University of California, Santa Barbara
Santa Barbara, CA 93106, USA
{maohuazhu, yuanxie}@ece.ucsb.edu

**Minsoo Rhu, Jason Clemons, Stephen W. Keckler**
NVIDIA Research
Austin, TX 78717, USA
{mrhu, jclemons, skeckler}@nvidia.com

## Abstract

Prior work has demonstrated that exploiting the sparsity can dramatically improve the energy efficiency and reduce the memory footprint of Convolutional Neural Networks (CNNs). However, these sparsity-centric optimization techniques might be less effective for Long Short-Term Memory (LSTM) based Recurrent Neural Networks (RNNs), especially for the training phase, because of the significant structural difference between the neurons. To investigate if there is possible sparsity-centric optimization for training LSTM-based RNNs, we studied several applications and observed that there is potential sparsity in the gradients generated in the backward propagation. In this paper, we investigate why the sparsity exists and propose a simple yet effective thresholding technique to induce further more sparsity during the LSTM-based RNN training. The experimental results show that the proposed technique can increase the sparsity of linear gate gradients to more than 80% without loss of performance, which makes more than 50% multiply-accumulate (MAC) operations redundant for the entire LSTM training process. These redundant MAC operations can be eliminated by hardware techniques to improve the energy efficiency and the training speed of LSTM-based RNNs.

## 1 Introduction

Deep neural networks have achieved state-of-the-art performance in many different tasks, such as computer vision (Krizhevsky et al., 2012) (Simonyan & Zisserman, 2015), speech recognition, and natural language processing (Karpathy et al., 2016). The underlying representational power of these neural networks comes from the huge parameter space, which results in an extremely large amount of computation operations and memory footprint. To reduce the memory usage and accelerate the training process, the research community has strived to eliminate the redundancy in the deep neural networks (Han et al., 2016b). Exploiting the sparsity in both weights and activations of Convolutional Neural Networks (CNNs), sparsity-centric optimization techniques (Han et al., 2016a) (Albericio et al., 2016) have been proposed to improve the speed and energy efficiency of CNN accelerators.

These sparsity-centric approaches can be classified into two categories: (1) pruning unimportant weight parameters and (2) skipping zero values in activations to eliminate multiply-accumulate (MAC) operations with zero operands. Although both categories have achieved promising results for CNNs, it remains unclear if they are applicable to training other neural networks, such as LSTM-based RNNs. The network pruning approach is not suitable for training because it only benefits the inference phase of neural networks by iteratively pruning and re-training. The approach that exploits the sparsity in the activations can be used for training because the activations are involved in both

the forward propagation and the backward propagation. But there are still some issues if we directly apply it to LSTM-based RNNs.

The sparsity in CNN activations mostly comes from the Rectified Linear Unit (ReLU) activation function, which sets all negative values to zero. However, Long Short-Term Memory, one of the most popular RNN cells, does not adopt the ReLU function. Therefore, LSTM should exhibit much less sparsity in activations than CNNs, intuitively. Furthermore, the structure of an LSTM cell is much more complicated than neurons in convolutional layers or fully connected layers of a CNN.

To explore additional opportunities to apply sparsity-centric optimization to LSTM-based RNNs, we conducted an application characterization on several LSTM-based RNN applications, including character-based language model, image captioning, and machine translation. Although the experimental results of the application characterization show that there is little sparsity in the activations, we observed potential sparsity in backward propagation of the LSTM training process. The activation values of the gates (input gate, forget gate, and output gate) and the new cell state exhibit a skewed distribution due to their functionality. That is, a large fraction of the activation values of these Sigmoid-based gates are either close to 1 or close to 0 (for the Tanh-based new cell activations, values are close to -1 or 1). This skewed distribution will lead to a considerable amount of very small values in the LSTM backward propagation since there is a term $\sigma(x)(1 - \sigma(x))$ in the gradients of the Sigmoid-based gates ($tanh(x)(1 - tanh(x))$ for the gradients of the new cell gradients), which will be zero given $\sigma(x) = 0$ or $\sigma(x) = 1$ ($tanh(x) = -1$ or $tanh(x) = 1$ for the new cell gradients). In real-world implementations, these very small values might be clamped to zero as they are in the form of floating-point numbers, of which the precision is limited. Therefore, there is potential sparsity in the gradients of the backward propagation of LSTM training.

To ensure that there is non-trivial amount of sparsity for hardware designers to exploit, we propose "sparsified" SGD, a rounding to zero technique to induce more sparsity in the gradients. This approach can be seen as a stochastic gradient descent (SGD) learning algorithm with sparsifying, which strips the precision of floating point numbers for unimportant small gradients. Experiment results show that with proper thresholds, we can make 80% of the gradients of the gate inputs to zero without performance loss for all applications and datasets we tested so far. As the sparse gradients of the gate inputs are involved in 67% matrix multiplications, more than 50% MAC operations are redundant in the entire LSTM training process. Eliminating these ineffectual MAC operations with hardware techniques, the energy efficiency and training speed of LSTM-based RNNs will be improved significantly.

## 2 BACKGROUND AND MOTIVATION

In this section, we first review some of the prior work on sparsity-centric optimization techniques for neural networks, and then illustrate the application characterization example as the motivation for our research.

### 2.1 SPARSITY-CENTRIC OPTIMIZATION FOR NEURAL NETWORKS

It has been demonstrated that there is significant redundancy in the parameterization of deep neural networks (Denil et al., 2013). Consequently, the over-sized parameter space results in sparsity in the weight parameters of a neural network. Besides the parameters, there is also sparsity in the activations of each layer in a network, which comes from two sources: (1) the sparsity in weight parameters and (2) the activation function of neurons, such as ReLU.

As the sparsity in weight parameters do not depend on the input data, it is often referred to as *static sparsity*. On the other hand, the sparsity in the activations depend on not only the weight values but also the input data. Therefore, we refer to the sparsity in the activations as *dynamic sparsity*.

Exploiting sparsity can dramatically reduce the network size and thus improve the computing performance and energy efficiency. For example, Deep Compression (Han et al., 2016b) applied network pruning to CNNs to significantly reduce the footprint of the weights, which enables us to store all the weights on SRAM. However, the static sparsity can only help the inference phase but not training because weight parameters are adjusted during training. Fortunately, leveraging the dynamic sparsity can benefit both inference and training of neural networks. Recent publications (Han et al.,

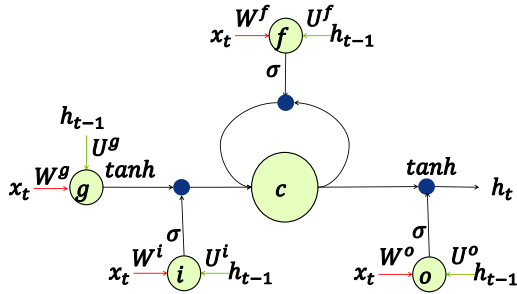

Figure 1: Basic LSTM cell

2016a) (Albericio et al., 2016) have proposed various approaches to eliminate ineffectual MAC operations with zero operands. Although these sparsity-centric optimization approaches have achieved promising results on CNNs, much less attention has been paid to LSTM-based RNNs, because there is a common belief that the major source of sparsity is the ReLU function, which is widely used in the convolutional layers but not in LSTM-based RNNs. To accelerate LSTM-based RNNs and improve the energy efficiency, we investigate opportunities to exploit sparsity in the LSTM-based RNN training process. As an initial step, in this paper we focus on the basic LSTM cell without peephole or other advanced features, as shown in Figure 1.

## 2.2 APPLICATION CHARACTERIZATION

To reveal if there is sparsity in LSTM training, we conduct an application characterization study. We start with a character-based language model as described in (Karpathy et al., 2016). This character-based language model takes a sequence of characters as input and predicts the next character of this sequence. The characters are represented in one-hot vectors, which are transformed into distributed vectors by a word2vec layer. Then the distributed vectors feed into an RNN model based on LSTM cells, followed by a linear classifier.

The LSTM cells used in this character-based language model are all basic LSTM cells. For each cell, the forward propagation flow is as below:

$$i_t = \sigma(W^i x_t + U^i h_{t-1} + b^i)$$
$$f_t = \sigma(W^f x_t + U^f h_{t-1} + b^f)$$
$$o_t = \sigma(W^o x_t + U^o h_{t-1} + b^o)$$
$$g_t = tanh(W^g x_t + U^g h_{t-1} + b^g)$$
$$c_t = f_t \circ c_{t-1} + i_t \circ g_t$$
$$h_t = o_t \circ tanh(c_t)$$

As shown in Figure 1, $i_t$, $f_t$, and $o_t$ stand for input gate, forget gate, and output gate, respectively. These sigmoid-based gates ($\sigma$ stands for sigmoid) are used to prevent irrelevant input from affecting the memory cell ($c_t$). The new cell state ($g_t$) is a preliminary summary of the current input from the previous layer and the previous status of current layer. The final hidden status $h_t$ is the output of the LSTM cell if it is seen as a black box.

Since the gates are introduced to prevent irrelevant inputs from affecting the memory cell $c_t$, we have a hypothesis that a large fraction of the activations of these gates should be either close to 1 or close to 0, representing the control signal on or off, respectively. Similarly, the tanh-based new cell status is active if its activation is 1 or inactive if it is -1. There should also be a considerable portion of the activations close to 1 or -1.

To validate our hypothesis, we extracted the activations of the sigmoid-based gates and tanh-based new cell state from several model snapshots during training the character-based language model. Figure 2 shows the histogram of the activation values of the gates and the new cell. The red curves represent the activation values generated by a snapshot model which is 0.5% trained (in terms of total number of iterations) while the bars represent the activation values generated by a fully trained

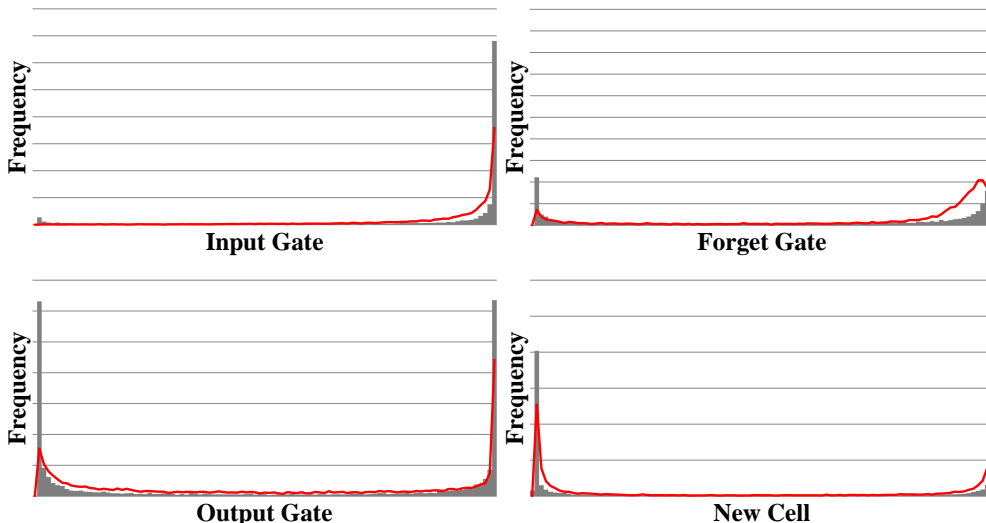

Figure 2: Values of gates and new cell activations of LSTM. For the three sigmoid-based gates, the range of x-axis is from 0 to 1. For the tanh-based new cell activation, the range is from -1 to 1.

model. We can observe skewed distributions from each gate (and new cell) for both the 0.5% trained snapshot model and the fully trained model. Furthermore, the fully trained model shows a distribution that is more skewed to the leftmost and the rightmost. Additionally, other un-shown snapshots demonstrate that the distribution becomes consistently more skewed as the training process goes on. We also observed that after 10% of the training process, the distribution becomes steady, almost the same as the fully trained model.

Besides the character-based language model, we also conducted the same characterization to the image captioning task described in (Karpathy & Li, 2015). The activation values of the RNN layer in the image captioning task exhibit the skewed distribution too. Even though we did not observe sparsity in the gate activations, the skewed distribution indicates potential sparsity in the LSTM-based RNN backward propagation, which will be shown in the next section.

## 3 SPARSIFIED STOCHASTIC GRADIENT DESCENT FOR LSTM

In this section, we first show how the skewed distribution of gate values leads to potential sparsity in the LSTM backward propagation, and then we propose the "sparsified" SGD to induce more sparsity in LSTM training.

### 3.1 POTENTIAL SPARSITY IN LSTM BACKWARD PROPAGATION

To show how the skewed distribution in the gate activations results in potential sparsity in the LSTM-based RNN backward propagation, we need to review the forward and backward propagation at first. We can re-write the forward propagation equations as

$$net(i)_t = W^i x_t + U^i h_{t-1} + b^i$$

$$net(f)_t = W^f x_t + U^f h_{t-1} + b^f$$

$$net(o)_t = W^o x_t + U^o h_{t-1} + b^o$$

$$net(g)_t = W^g x_t + U^g h_{t-1} + b^g$$

$$i_t = \sigma(net(i)_t)$$

$$f_t = \sigma(net(f)_t)$$

$$o_t = \sigma(net(o)_t)$$

$$g_t = tanh(net(g)_t)$$

$$c_t = f_t \circ c_{t-1} + i_t \circ g_t$$

$$h_t = o_t \circ tanh(c_t)$$

Here we introduce variables $net(i)$, $net(f)$, $net(o)$ and $net(g)$ to represent the linear part of the gates and the new cell state. In GPU implementations such as cuDNN v5 (Appleyard et al., 2016), these linear gates (including new cell state from now on) are usually calculated in one step since they share the same input vectors $x_t$ and $h_{t-1}$. Therefore we can use a uniform representation for the four linear gates, that is

$$net_t = Wx_t + Uh_{t-1} + b$$

The matrix $W$ here stands for the combination of the matrices $W^i$, $W^f$, $W^o$ and $W^g$ and the matrix $U$ stands for the combination of the matrices $U^i$, $U^f$, $U^o$ and $U^g$.

With these denotations, we can express the backward propagation as

$$do_t = dh_t \circ tanh(c_t)$$

$$dc_t = dh_t \circ (1 - tanh^2(c_t)) \circ o_t + f_t \circ c_{t+1}$$

$$dnet(g)_t = dc_t \circ i_t \circ (1 - g_t^2)$$

$$dnet(o)_t = dh_t \circ tanh(c_t) \circ (1 - o_t) \circ o_t$$

$$dnet(f)_t = dc_t \circ c_{t-1} \circ (1 - f_t) \circ f_t$$

$$dnet(i)_t = dc_t \circ g_t \circ (1 - i_t) \circ i_t$$

$$dx_t = dnet_t W^T$$

$$dh_{t-1} = dnet_t U^T$$

$$dW += x_t dnet_t$$

$$dU += h_{t-1} dnet_t$$

In the equations of the backward propagation, we use $dnet$ to denote the gradient of the linear gates.

From these equations we can see that for each linear gate gradient there is one term introduced by the sigmoid function or the tanh function, e.g. $(1 - g_t^2)$ in $dnet(g)_t$ and $(1 - o_t) \circ o_t$ in $dnet(o)_t$. As we observed in the application characterization results, the activation values of these gates exhibit skewed distribution, which means a large fraction of $o_t$, $f_t$ and $i_t$ are close to 0 or 1 ($g_t$ close to -1 or 1). The skewed distribution makes a large fraction of the linear gate gradients close to zero because $(1 - g_t^2)$, $(1 - o_t) \circ o_t$, $(1 - f_t) \circ f_t$ and $(1 - i_t) \circ i_t$ are mostly close to zero given the skewed distribution of the gate activations.

When implementing the LSTM-based RNNs, we usually use 32-bit floating point numbers to represent the gradients. Due to the precision limit, floating point numbers will round extremely small values to zero. Therefore, there is potential sparsity in $dnet$ since a large fraction of the linear gate gradients are close to zero.

## 3.2 Inducing More Sparsity

In the previous section we showed how the skewed distribution in gate activations results in potential sparsity in linear gate gradients theoretically. However, from mathematical perspective, there will be no sparsity in linear gate gradients if the floating point numbers in computers have infinite precision since they are only close to zero rather than be zero. Even the precision of 32-bit floating point numbers is not infinite, the 8-bit exponential part can still accommodate an extremely large dynamic range, which makes the sparsity less interesting to hardware accelerator designers. Fortunately, recent attempts to train neural networks with 16-bit floating points (Gupta et al., 2015) and fixed points (Lin et al., 2015) have shown acceptable performance with smaller dynamic range. This inspires us to induce more sparsity by rounding very small linear gate gradients to zero, which is similar to replace 32-bit floating points with 16-bit floating points or fixed points.

The intuition behind this "*rounding to zero*" approach is that pruning CNNs will not affect the overall training performance. Similarly, thresholding very small gradient ($dnet$) values to zero is likely not to affect the overall training accuracy. Therefore, we propose a simple static thresholding approach which sets small $dnet$ values below a threshold $t$ to zero. By doing this, we can increase the sparsity in $dnet$ even further than the original sparsity caused by limited dynamic range of floating

point numbers. With our static thresholding technique, the backward propagation of LSTM training becomes as below:

$$do_t = dh_t \circ tanh(c_t)$$
$$dc_t = dh_t \circ (1 - tanh^2(c_t)) \circ o_t + f_t \circ c_{t+1}$$
$$dnet(g)_t = dc_t \circ i_t \circ (1 - g_t^2)$$
$$dnet(o)_t = dh_t \circ tanh(c_t) \circ (1 - o_t) \circ o_t$$
$$dnet(f)_t = dc_t \circ c_{t-1} \circ (1 - f_t) \circ f_t$$
$$dnet(i)_t = dc_t \circ g_t \circ (1 - i_t) \circ i_t$$
$$\mathbf{dnet_t = (dnet_t > t)?dnet_t : 0}$$
$$dx_t = dnet_t W^T$$
$$dh_{t-1} = dnet_t U^T$$
$$dW+ = x_t dnet_t$$
$$dU+ = h_{t-1} dnet_t$$

In this "sparsified" SGD backward propagation, a new hyper-parameter $t$ is introduced to control the sparsity we would like to induce in $dnet$. Clearly, the optimal threshold $t$ is the highest one that has no impact on the training performance since it can induce the highest sparsity in $dnet$. Therefore, to select the threshold, we need to monitor the impact on the gradients. As the SGD only uses the gradients of the weights ($dW$) to update the weights, $dW$ is the only gradients we need to care about. From the equations of the backward propagation we can see that $dW$ is computed based on $dnet$, which is sparsified by our approach. Although sparsifying $dnet$ affects $dW$, we can control the change of $dW$ by setting the threshold. To determine the largest acceptable threshold, we conducted an evaluation of the impact caused by different thresholds on one single step in LSTM training. The application here is the same as the one in the application characterization.

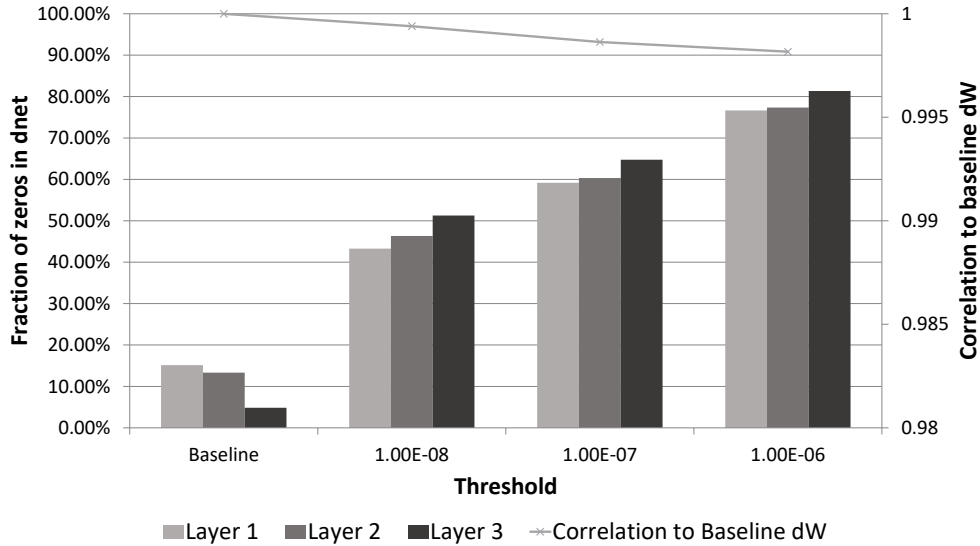

Figure 3: Impact of Training with Sparsified SGD

Figure 3 shows the evaluation result. We measure the change of $dW$ by the normalized inner product of sparsified $dW$ and the original $dW$ without sparisifying (the baseline shown in Figure 3). If we denote the original weight gradient as $dW_0$, the correlation between sparsified $dW$ and $dW_0$ can be measured by normalized inner product

$$correlation = \frac{dW \cdot dW_0}{||dW|| \cdot ||dW_0||}$$

If the correlation is 1, it means $dW$ is exactly the same to $dW_0$. If the correlation is 0, it means $dW$ is orthogonal to $dW_0$. The higher the correlation is, the less impact the sparsification has on this single step backward propagation. From Figure 3 we can see that even without our thresholding technique, the $dnet$ still exhibits approximately 10% sparsity. These zero values are resulted from the limited dynamic range of floating point numbers, in which extremely small values are rounded to zero. By applying the thresholds to $dnet$, we can induce more sparsity shown by the bars. Even with a low threshold ($10^{-8}$), the sparsity in $dnet$ is increased to about 45%. With a relatively high threshold ($10^{-6}$), the sparsity can be increased to around 80%. Although the sparsity is high, the correlation between the sparsified $dW$ and $dW_0$ is close to 1 even with the high threshold. Therefore, we can hypothesize that we can safely induce a considerable amount of sparsity with an appropriate threshold. It is straightforward to understand that the threshold cannot be arbitrarily large since we need to contain the information of the gradients. For example, if we increase the threshold even further to $10^{-5}$, the correlation will drop to 0.26, which is far from the original $dW_0$ and not acceptable.

We have demonstrated that we can induce more sparsity by rounding small $dnet$ to zero while maintaining the information in $dW$. However, this is only an evaluation on one single iteration of training. To show the generality of our static thresholding approach, we applied the thresholds to the entire training process.

## 4 AN ENTIRE LSTM TRAINING WITH SPARSIFIED SGD

In this section, we first present the sparsity induced by applying our sprsified SGD to an entire training process, and then discuss the generality of our approach.

### 4.1 CHARACTER-BASED LANGUAGE MODEL

To validate our proposed static thresholding approach, we apply it to the entire LSTM-based RNN training process. We first conducted an experiment on training a character-based language model. The language model consists of one word2vec layer, three LSTM-based RNN layers, and one linear classifier layer. The number of LSTM cells per RNN layer is 256. We feed the network with sequences of 100 characters each. The training dataset is a truncated Wikipedia dataset. We apply a fixed threshold to all $dnet$ gradients for every iteration during the whole training process.

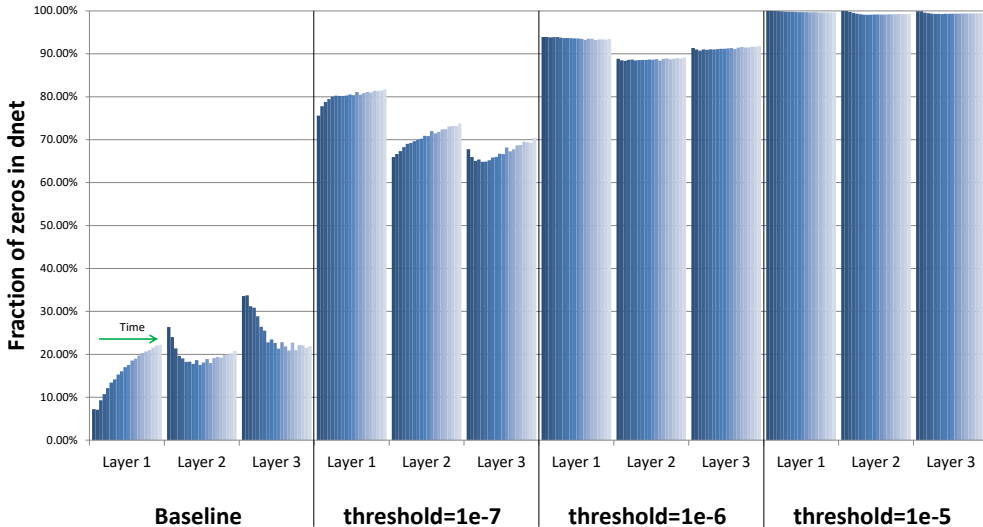

Figure 4: Sparsity in dnet with different thresholds

Figure 4 shows the sparsity of the linear gate gradients ($dnet$) of each layer during the whole training process. In the baseline configuration, the training method is standard SGD without sparsifying (zero

threshold). The baseline configuration exhibits about 20% sparsity in $dnet$. By applying only a low threshold ($10^{-7}$), the sparsity is increased to around 70%. And we can consistently increase the sparsity further by raising the threshold. However, we have to monitor the impact of the threshold on the overall training performance to check if the threshold is too large to use.

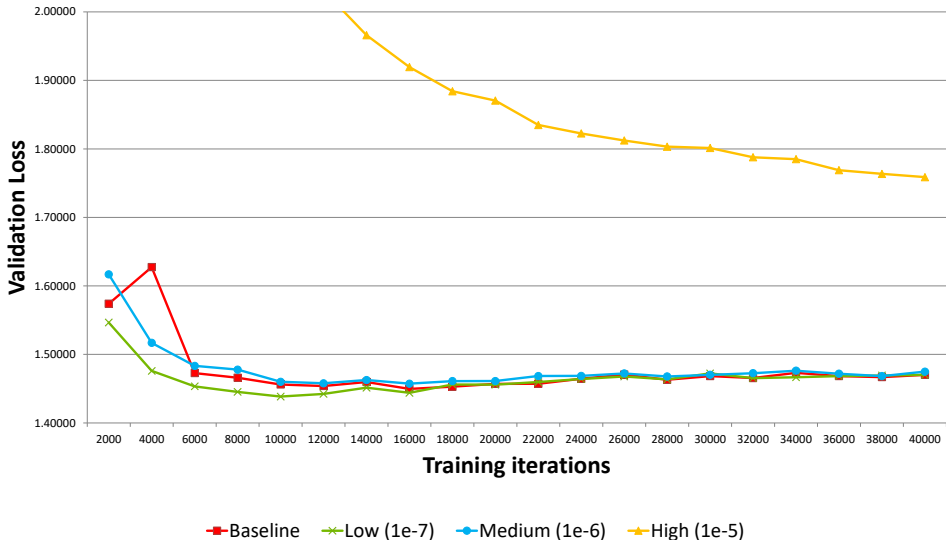

Figure 5: Validation Loss with different thresholds

Figure 5 shows the validation loss of each iteration. We observe that up to the medium threshold ($10^{-6}$), the validation loss of the model trained with sparsified SGD keeps close to the baseline. However, if we continues raising the threshold to $10^{-5}$, the validation loss becomes unacceptably higher than the baseline. Although the validation loss with the $10^{-5}$ threshold is consistently decreasing as the training goes on, we conservatively do not pick this configuration to train the LSTM network. So combining Figure 4 with Figure 5, we can choose the threshold $10^{-6}$ to train the character-based language model to achieve about 80% sparsity in $dnet$.

Since the linear gate gradients $dnet$ are involved in all the four matrix multiplications in the backward propagation, there are 80% MAC operations in these matrix multiplications have zero operands. Furthermore, there are six matrix multiplications (all of them are of the same amount of computation) in one LSTM training iteration and four out of them (67%) are sparse. So there are more than 50% MAC operations will have zero operands introduced by our sparsified SGD in one LSTM training iteration. The MAC operations with zero operands produce zero output and thus make no contribution to the final results. These redundant MAC operations can be eliminated by hardware techniques similar to (Han et al., 2016a) (Albericio et al., 2016) to improve the energy efficiency of LSTM training.

## 4.2 SENSITIVITY TEST

Our static thresholding approach can induce more than 80% sparsity in linear gate gradients of the character-based language model training. To demonstrate the generality of our approach, we then changed the topology of the RNN layers in the character-based language model with several different LSTM-based RNNs for a sensitivity test. The network topologies used in the sensitivity test are shown below.

- Number of layers: 2, 3, 6, 9;
- Number of LSTM cells per layer: 128, 256, 512;
- Sequence length: 25, 50, 100.

We also trained the network with other datasets, such as the tiny-Shakespear dataset and the novel War and Peace. For all the data points we collected from the sensitivity test, we can always achieve

more than 80% sparsity in $dnet$ with less than 1% loss of performance in terms of validation loss with respect to the baseline.

Moreover, we also validated our approach by training an image captioning application (Karpathy & Li, 2015) with MSCOCO dataset (Lin et al., 2014) and a machine translation application known as Seq2Seq (Sutskever et al., 2014) with WMT15 dataset. As both the two applications are implemented based on graph model (Torch and TensorFlow, respectively), we plugged a custom operation in the automatically generated backward propagation subgraph to implement our proposed sparsified SGD. The experiment results show that the conclusion for the character-based language model still holds for the two applications.

## 4.3 DISCUSSION

So far all our experiment results show promising results and we believe our sparsified SGD is a general approach to induce sparsity for LSTM-based RNN training. From the computer hardware perspective, the sparsified SGD is similar to reduced precision implementation while the impact of sparsified SGD is much less since we still use full 32-bit floating point numbers. From the theory perspective, SGD itself is a gradient descent with noise and thresholding very small gradients to zero is nothing more than an additional noise source. Since training with SGD is robust to noise, the thresholding approach will likely not affect the overall training performance. Additionally, the weight gradients $dW$ are aggregated through many time steps, which makes the LSTM more robust to the noise introduce by sparsifying the linear gate gradients.

## 5 CONCLUSION AND FUTURE WORK

In this paper, we conducted an application characterization to an LSTM-based RNN application and observe skewed distribution in the sigmoid-based gates and the tanh-based new cell state, which indicates potential sparsity in the linear gate gradients during backward propagation with SGD. The linear gate gradients are involved with 67% MAC operations in an entire LSTM training process so that we can improve the energy efficiency of hardware implementations if the linear gate gradients are sparse. We propose a simple yet effective rounding to zero technique, which can make the sparsity of the linear gate gradients higher than 80% without loss of performance. Therefore, more than 50% MAC operations are redundant in an entire sparsified LSTM training.

Obviously, the static-threshold approach is not optimal. In future, we will design a dynamic-threshold approach based on the learning rate, L2-norm of the gradients and the network topology. Hardware techniques will also be introduced to exploit the sparsity to improve the energy efficiency and training speed of LSTM-based RNNs for GPU and other hardware accelerators.

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
