# Peer review of "Training Long Short-Term Memory With Sparsified Stochastic Gradient Descent"

_ICLR 2017 — rejected_

[Official Review · AnonReviewer2 · rating 4 · confidence 4 · 16 Dec 2016]
**Review: Training Long Short-Term Memory with Sparsified Stochastic Gradient Descent**

CONTRIBUTIONS
When training LSTMs, many of the intermediate gradients are close to zero due to the flat shape of the tanh and sigmoid nonlinearities far from the origin. This paper shows that rounding these small gradients to zero results in matrices with up to 80% sparsity during training, and that training character-level LSTM language models with this sparsification does not significantly change the final performance of the model. The authors argue that this sparsity could be exploited with specialized hardware to improve the energy efficiency and speed of recurrent network training.

NOVELTY
Thresholding gradients to induce sparsity and improve efficiency in RNN training is a novel result to my knowledge.

MISSING CITATIONS
Prior work has explored low-precision arithmetic for recurrent neural network language models:

Hubara et al, “Quantized Neural Networks: Training Neural Networks with
Low Precision Weights and Activations”,

[Official Review · AnonReviewer1 · rating 4 · confidence 4 · 17 Dec 2016]
**No Title**

The findings of applying sparsity in the backward gradients for training LSTMs is interesting. 

But the paper seems incomplete without the proper experimental justification. Only the validation loss is reported which is definitely insufficient. Proper testing results and commonly reported evaluation criterion needs to be included to support the claim of no degradation when applying the proposed sparsity technique. 

Also actual justification of the gains in terms of speed and efficiency would make the paper much stronger.

[Official Review · AnonReviewer3 · rating 5 · confidence 3 · 17 Dec 2016]
**Detailed analysis/implementation needed**

This paper presents the observation that it is possible to utilize sparse operations in the training of LSTM networks without loss of accuracy. This observation is novel (although not too surprising) to my knowledge, but I must state that I am not very familiar with research on fast RNN implmentations.

Minor note:
The LSTM language model does not use a 'word2vec' layer. It is simply a linear embedding layer. Word2vec is the name of a specific model which is not directly to character level language models.

The paper presents the central observation clearly. However, it will be much more convincing if a well known dataset and experiment set up are used, such as Graves (2013) or Sutskever et al (2014), and actual training, validation and test performances are reported.

While the main observation is certainly interesting, I think it is not sufficient to be the subject of a full conference paper without implementation (or simulation) and benchmarking of the promised speedups on multiple tasks. For example, how would the gains be affected by various architecture choices?

At present this is an interesting technical report and I would like to see more detailed results in the future.

[Final Decision · Program Chairs · 06 Feb 2017]
**ICLR committee final decision**

The main point of the paper was that sparsifying gradients does not hurt performance; however, this in itself is not enough for a publication in this venue. As noted by R1 and R2, showing how this can help in more energy efficient training would make for a good paper; without that aspect the paper only presents an observation that is not too surprising to the practitioners in this area.
 
 Further, while the main point of the paper was relatively clear, the scientific presentation was not rigorous enough. All the reviewers point out that several details were missing (including test set performance, reporting of results on the different sets, etc). 
 
 Paper would be strengthened by a better exploration of the problem.